# Advances in and the Applicability of Machine Learning-Based Screening and Early Detection Approaches for Cancer: A Primer

**DOI:** 10.3390/cancers14030623

**Published:** 2022-01-26

**Authors:** Leo Benning, Andreas Peintner, Lukas Peintner

**Affiliations:** 1Health Care Supply Research and Data Mining Working Group, Emergency Department, University Medical Center Freiburg, 79106 Freiburg, Germany; leo.benning@uniklinik-freiburg.de; 2Databases and Information Systems, Department of Computer Science, Leopold-Franzens University of Innsbruck, 6020 Innsbruck, Austria; andreas.peintner@uibk.ac.at; 3Institute of Molecular Medicine and Cell Research, Albert Ludwigs University of Freiburg, 79085 Freiburg, Germany

**Keywords:** cancer diagnostics, machine learning, artificial intelligence, high throughput, deep learning, CNN, DNN

## Abstract

**Simple Summary:**

Non-communicable diseases in general, and cancer in particular, contribute greatly to the global burden of disease. Although significant advances have been made to address this burden, cancer is still among the top drivers of mortality, second only to cardiovascular diseases. Consensus has been established that a key factor to reduce the burden of disease from cancer is to improve screening for and the early detection of such conditions. To date, however, most approaches in this field relied on established screening methods, such as a clinical examination, radiographic imaging, tissue staining or biochemical markers. Yet, with the advances of information technology, new data-driven screening and diagnostic tools have been developed. This article provides a brief overview of the theoretical foundations of these data-driven approaches, highlights the promising use cases and underscores the challenges and limitations that come with the introduction of these approaches to the clinical field.

**Abstract:**

Despite the efforts of the past decades, cancer is still among the key drivers of global mortality. To increase the detection rates, screening programs and other efforts to improve early detection were initiated to cover the populations at a particular risk for developing a specific malignant condition. These diagnostic approaches have, so far, mostly relied on conventional diagnostic methods and have made little use of the vast amounts of clinical and diagnostic data that are routinely being collected along the diagnostic pathway. Practitioners have lacked the tools to handle this ever-increasing flood of data. Only recently, the clinical field has opened up more for the opportunities that come with the systematic utilisation of high-dimensional computational data analysis. We aim to introduce the reader to the theoretical background of machine learning (ML) and elaborate on the established and potential use cases of ML algorithms in screening and early detection. Furthermore, we assess and comment on the relevant challenges and misconceptions of the applicability of ML-based diagnostic approaches. Lastly, we emphasise the need for a clear regulatory framework to responsibly introduce ML-based diagnostics in clinical practice and routine care.

## 1. Introduction

The global burden of disease from non-communicable diseases has been on the rise for decades. Among other factors, tobacco use, physical inactivity, unhealthy diets and the excessive consumption of alcohol have greatly contributed to this trend [1]. The latest Global Burden of Disease report of The Lancet, for example, identifies six non-communicable diseases, among the ten top drivers of the global burden of disease [2].

Among these, malignant diseases are of particular importance. Although major age-standardised indicators show a significant decline in, for example, cancer mortality since the 1990s [3], the global trends imply that the goal of eliminating cancer as one of the major health concerns has not been achieved yet. The global burden of cancer was estimated to account for 23.6 million cases in 2019, excluding non-melanoma skin cancer, and ten million deaths in the same year [4]. Even in the U.S., with a highly developed and high-resourced health care system, only about half of the patients with cancer could be cured with existing therapies in 2019, while the remaining half was expected to die with or of their disease [5].

The challenges that arise with these trends have long been recognised on a supra-national level. As part of the Sustainable Development Goals (SDGs), the United Nations (UN) has defined a global target to reduce the impact of non-communicable diseases among premature populations (i.e., between the age of 30 and 70) by 30%. According to recent analyses, the achievement of this reduction would not only result in an expanded working population, a higher productivity and reduced disparity, but it would also yield significant social, economic and political returns [6]. The key drivers to achieve this reduction have been identified accordingly and include, among surveillance, prioritised prevention and improved therapies, also the screening for and the early detection of non-communicable diseases in general and malignant conditions in particular.

It has become clear, and it is global consensus now, that the diagnosis of cancers in early stages of the disease holds great potential to effectively reduce the burden of disease and to facilitate an early treatment initiation that typically yields a better prognosis for the individual patients [7]. Well-designed and thoroughly implemented screening programs have already led to a significant reduction of burden of disease already, but also come with relevant drawbacks [8]. Population-wide screening programs that demonstrated relevant positive outcomes, however, remain relatively sparse, being mostly limited to image-based breast cancer, more recently, lung cancer screening, cytology-based cervical cancer screening, clinical, endoscopic and histological colorectal cancer screening, as well as clinical and biochemical prostate cancer screening [3]. All of these approaches have in common that risks of overdiagnosis and overtreatment can not only be harmful for the individual patient, but also for the system that employs these screening programs [9]. This debate has been particularly prominent around the prostate-specific antigen (PSA) screening for prostate cancer [10]. This highlights that most approaches toward screening and early detection are still based on conventional diagnostic methods, such as clinical examination, cytology, histology, radiographic imaging or biochemical tests. However, the advancement of computer technology and new approaches in machine learning (ML) ushers in an entirely new field of diagnostic approaches that bear the potential to address the challenges of screening and early detection very effectively [11]. The impact of ML on cancer diagnostics will be the focus of this article, which aims to summarise (a) the status quo of the screening for and early detection of malignant diseases using ML, (b) identify the challenges and opportunities within the field and (c) contrast these with the gold standard approaches for the detection of malignant diseases. Since, across both sexes, female breast cancer has recently become the most common type of cancer, accounting for about 11.7% of all cancer diagnoses, which is followed by lung cancers (11.4%), colorectal cancers (10%) and prostate cancers (7.3%) [7], this review will primarily focus on these entities and common the aspects of their respective diagnostic funnels.

## 2. A Brief Introduction to Machine Learning

Artificial intelligence (AI) is an umbrella term and is used to describe so-called intelligent agents, which are able to perceive external hazards and perform corresponding actions to reach their goal [12]. A specific branch of artificial intelligence is machine learning (ML). This branch uses mathematical and statistical techniques to learn from data and perform inference. This learning process is subdivided into two phases: in the first phase, the model estimates the unknown dependencies in the given data, followed by the second phase in which the model uses these learned dependencies to predict new outputs [13]. Due to the increasing availability of data and better hardware capabilities, ML has become a very powerful tool in various areas of research [14,15,16].

In ML, different tasks can be designed and allow the application of certain methods in a desired framework: the supervised learning task requires a labelled set of training data to estimate a function that maps the input data to the given output value. In a regression problem, the task is to predict a real-value variable from the data and can be seen as a special case of supervised learning. Example algorithms for supervised learning include support-vector machines (SVMs), decision trees (DTs), k-nearest neighbour (kNN) and neural networks (NNs). In contrast to the supervised learning tasks, the unsupervised learning task does not require labelled training data and so the model is agnostic of any output during the learning process. Consequently, the task of the model centres on the discovery of any patterns or groups in the input data. Clustering, as an example of an unsupervised learning task, automatically groups the training inputs into clusters of similar features. New samples are assigned to a cluster according to similar characteristics. A third widely used ML task is semi-supervised learning, which combines labelled and unlabelled data to provide more accurate predictions [13].

### 2.1. Overview of Deep Learning Methods

Deep learning is a class of ML methods and is inspired by the information processing in biological systems. These models use multiple stacked layers (therefore “deep”) with numerous neurons per layer to progressively extract features from the input data. Current state-of-the-art methods in image recognition and natural language processing are built upon so-called seep neural networks (DNNs) and mostly outperform traditional ML approaches [17,18].

Various applications of deep learning adopt feed-forward neural networks (NNs), which are able to map a fixed-length input to a fixed-length output. The basic building blocks of feed-forward neural networks are neurons (alternatively called units). Each neuron computes a weighted sum of its inputs and applies a non-linear activation function to it. A commonly used activation function is the rectified linear unit (ReLU) function, which is defined as:(1)$$f\left(z\right)=\max\left(z,0\right)$$

A feed-forward neural network consists of an input layer, one or more hidden layers and an output layer. The output of one layer acts as an input for the next, sequentially following layer, as can be seen in Figure 1a.

To train such a deep neural network, a loss function is minimised by an optimisation algorithm. The backpropagation procedure calculates the gradient of a loss function with respect to the weights of a network by applying the chain rule of derivatives. This is used to propagate all the gradients back to the input layer, where it is then used to compute the gradients with respect to the weights of each neuron. An optimisation algorithm, such as stochastic gradient descent (SGD), then updates the weights and biases based on only one sample from the dataset (hence stochastic) and its partial derivatives with respect to the parameters per iteration [22].

A popular approach for training the deep learning models is transfer learning. In the transfer learning scenario, a pre-trained model based on a different but related task is used as a starting point for fine-tuning on the actual data. The implicit knowledge contained in the pre-trained model supports the learning process and improves the performance of the fine-tuned model.

#### 2.1.1. Multilayer Perceptron (MLP)

Multilayer perceptrons are a special case of feed-forward neural networks and a fundamental architecture for deep learning approaches. This prediction function is composed by multiple fully connected (FC) layers, including input and output layers, and an activation function (for example, ReLU or sigmoid function) σ :ℝ→ℝ, which is applied to each component as seen in:(2)yˇ=WkσWk−1σ…σW1χ+b1…+bk−1+bk
where Wl∈ℝnl×nl−1 is the weight matrix and bl∈ℝnl is the bias in layer l=1,…,k. It is shown that MLPs are able to fit data with arbitrary precision, given enough parameters in terms of the depth and width. Therefore, MLPs can be seen as universal function approximators and are able to capture more complex interactions [23].

#### 2.1.2. Convolutional Neural Network (CNN)

Feed-forward neural networks and multilayer perceptrons are commonly used for structured data, whereas convolutional neural networks are mostly applied to image data. These networks contain one or more convolutional layers, where each layer multiplies a kernel (i.e., a set of learnable parameters) with the portion of the receptive field in the image. This results in an activation map, which reflects the response of the kernel at different spatial positions.

Convolutions are linear operations, therefore we usually introduce non-linearity in the resulting activation maps by non-linear layers with, e.g., ReLU as an activation function. To reduce the computational complexity, pooling subsamples neighbourhood outputs and replaces the outputs by its summary statistics. Max-pooling is a commonly used pooling function which simply returns the maximum value of the neighbourhood (Figure 1b). Lastly, all these layers end up in one or more fully connected layers, which map the learned representations of the input image to the output [24,25].

Stacking many layers leads to the degradation problem, where accuracy is prone to saturate and degrade with the increasing depth of the network. Deep residual learning, as introduced in [17], tackles this problem via shortcut or skip connections. These connections allow passing feature information on to the lower layers. This approach led to the widely used ResNet [17] architecture. Other common architectures include AlexNet [14], VGGNet [26], GoogleNet [27] and SqueezeNet [28].

The input images for CNNs usually come in 2D, but for medical use cases volumetric images are often the norm. A 3D-CNN is the counterpart to a 2D-CNN, which takes a sequence of 2D frames as input. The convolution and pooling layers are adopted according to the altered input format [29].

#### 2.1.3. Fully Convolutional Network (FCN)

These kinds of networks are used for semantic segmentation, where each pixel in the image is classified and therefore results in the segmented maps of objects. In comparison to CNNs, FCNs drop the fully connected layers and append up-sampling and deconvolutional layers to reverse the proceeding convolution and pooling operations [30].

An adaptation of the FCN architecture in the area of biomedical image segmentation is called a U-Net [31]. They optimise the original FCN for small amounts of training images and more precise segmentations. Feature channels in the up-sampling part enables the network to pass-through context information to higher layers (Figure 1c).

#### 2.1.4. Recurrent Neural Network (RNN)

Recurrent neural networks (RNNs) are able to exhibit temporal dynamic behaviour and are therefore often used for text and sequential data in general. In these networks, information of the preceding elements of the sequence have an effect on the current input and output [32]. To be more formal, given an input sequence $x=\left(x_{1},\ldots ,x_{T}\right)$, the hidden sequence vector and the output sequence is computed in the following way:(3)$$h_{t}=\sigma \left(W_{xh}x_{t}+W_{hh}h_{t-1}+b_{h}\right)\;y_{t}=W_{hy}h_{t}+b_{y}$$
with *W* as the weight matrix, *b* as the bias vector and *σ* as the hidden layer function (e.g., sigmoid).

As opposed to the unidirectional RNNs, bidirectional recurrent neural networks (BRNNs) are enabled to use future information of the sequence to improve the performance of the model (Figure 1d). Another form of RNN architecture is the long short-term memory (LSTM) model, which addresses the problem of long-term dependencies by custom memory cells to utilise long-range context in the sequence [33]. Gated recurrent units (GRUs), as introduced in [34], similarly address the issue of the short-term memory of RNNs by so-called reset and update gates, which control the information retention.

The combination of CNNs and RNNs leads to convolutional recurrent neural networks (CRNNs) [35]. In this architecture, the CNN part extracts the features from the input images and the recurrent neural network then processes these features in order and learns the possible links between those features.

## 3. Machine Learning Applications in Cancer Diagnostics

Commonly, the diagnostic pathway for a cancerous condition is typically initiated through a routine screening in the asymptomatic patient, or through the active consultation of a somewhat symptomatic patient. Based on the correct interpretation of the diagnostic results obtained and the diagnostic workup for the respective condition, the practitioner then decides the treatment regimen for the patient, typically based on national or international guidelines. Since clinical data, mostly in the form of digital images or structured data, are easily machine readable and since the path to a qualified diagnosis is, generally speaking, strictly logical, ML approaches are well suited to support the medical staff to instrumentalise the vast amounts of diagnostic information more efficiently. However, due to the diverse nature of cancer and the multitude of processed information a practitioner may ask for, there is no one-fits-it-all algorithm that can serve all tasks. This chapter introduces the standard ML algorithms that evolved over the last decade and were employed for specific diagnostic tasks and data (Table 1).

To understand and interpret diagnostic data, one needs to perform a series of analysis steps, which evolved into different tasks addressed by ML algorithms.

The simplest readout comes with detection algorithms. Detection identifies tumorous tissue in a binary decision model, typically based on radiographic imaging or histological samples.

A little more information will be provided by classification. This algorithms identify tumour tissues and enable the qualitative analysis and potential implications of this tumour for the patient. It contains the identification of objects of different types and multilabel classifications.

A more sophisticated outcome is accomplished with segmentation. This process identifies and localises (e.g., mark area on an image) potential malignant tissue from healthy intact tissue. This task is performed on all kinds of visual data, such as whole slide images (WSIs), mammograms, MRI and CT scans [36].

When it comes to the analysis of OMICs data, the nature of the data completely changes. OMICs data are the expression data of biochemical molecules, such as RNA, proteins or metabolites [11]. A comparison of expression patterns can be used to perform prognosis prediction.

Over the last several years, ML algorithms emerged that are able to support the clinician in the interpretation of data (Table 1).

Several seminal and influential publications were published over the past decade. While earlier works in the field are thoroughly reviewed in [13], more recent advances were comprehensively reviewed in [24,25,66]. This article, however, focuses on the primary works highlighted in Table 1, as we deem the examples provided as particularly insightful to understand the (a) theoretical foundations and the (b) clinical applications of each approach outlined. Applying ML in breast cancer diagnosis produced various approaches on different input data types. Work in [41] first classifies an input mammogram for the inclusion of potential masses via a CNN classifier. An FCN based on this CNN is then used to output probability regions to localise the mass in the image. Similarly, in [58], a deep learning detector is trained to detect breast lesions from entire mammograms. In a second step, different state-of-the-art CNN classification models predict the mass to be benign or malignant. A custom CNN architecture introduced in [39] recognises the abnormal mammogram images. In [58], transfer learning is used to alleviate the need for large datasets. Image features are extracted via a CNN and are then analysed by a SVM classifier. Ultrasound images are another widely used input data type for breast cancer diagnosis with ML. Based on the introduced U-Net architecture, the approach in [42] segments the suspicious breast masses on ultrasound images and is able to output predictions in real time (13–55 ms per image). Several transfer learning techniques in combination with a matching layer performing colour conversion to efficiently utilise the pretrained model are applied in [49]. The CNN based on the VGG neural network architecture was pre-trained on the ImageNet dataset and then fine-tuned on a dataset of 882 ultrasound images of breast masses. In [67], GoogleNet was trained to distinguish between benign and malignant tumours. A custom dataset was built containing 7408 ultrasound breast images. Augmentation techniques to artificially increase the dataset size included histogram equalisation, image cropping and margin augmentation. The approach in [44], combines multiple fine-tuned CNN models (including ResNet) in a so-called ensemble model to classify histopathological microscopy images into normal tissue, benign lesion, in situ carcinoma and invasive carcinoma. A custom technique named DNNS, enhances the input images by particular preprocessing steps and features extraction methods to improve the performance of different classifier models in [45].

Based on the MRI data, which congenitally comes in a 3D format, a multi-scale 3D-CNN is introduced in [38] to accurately segment the brain lesions. A second pathway that operates on down-sampled images is added to the model to effectively learn high level features, such as the location in the brain. The detailed local appearance of the lesion is captured in the first pathway, which operates on default scaled images. The methylation status of MGMT indicates the response to temozolomide and possibly influences the overall patient survival. Work in [57] utilised different ResNet architectures to predict the MGMT methylation status on the basis of the MRI data. Similarly, in [19], a bi-directional convolutional recurrent neural network architecture (CRNN) is used to predict the methylation state of the MGMT regulatory regions. In [38], MR images are used to automatically segment the prostate by extracting features via a CNN to generate a set of prostate proposals. For each of these proposals, a graph is generated based on the extracted features to finally select the best prostate proposal segmentation.

Classifying small lung nodules as malignant is clinically difficult, since they cannot be reliably characterised. The authors in [50] tackled this task by introducing a deep belief network (DBN) model and a CNN model trained on computed tomography (CT) images for nodule classification. Another approach is presented in [51], where features from a pre-trained CNN (AlexNet) model are combined with hand-crafted features to train a random forest (RF) classifier. TumorNet, as introduced in [52], projects the CT scans into 2D patches for each dimension and concatenates them. Each dimension corresponds to the different channels of the input image for the CNN. A Gaussian process (GP) regression is performed to predict the malignancy score of a nodule. The 3D-CNN used in [59] fully exploits the input format of CT scans by extracting richer spatial information and more representative features, based on the hierarchical architecture for pulmonary nodule detection.

The correct differentiation between the benign and malignant skin lesions is crucial to subsequent patient observation and treatment. In [46], a hybrid deep learning approach is presented. The extracted features from pre-trained AlexNet, VGG16 and ResNet-18 are used to train separate SVM classifiers. The outputs of the classifiers are fused in the final stage to obtain the predicted class of the input dermoscopic image. In Deepmole [47], features extracted from the last three layers of the introduced CNN architecture are used by a k-nearest neighbour classifier (kNN) to differ between melanoma and benign samples. The availability of haematoxylin–eosin (H&E) stained tissue slides of colorectal cancer enabled the application of deep learning algorithms to extract the prognosticators for diagnosis. In [56], several CNN architectures, including AlexNet, VGG19, ResNet-50, SqueezeNet and GoogleNet, were evaluated on 862 H&E slides for predicting nine possible output classes.

A combination of the classification and prediction of the outcomes is established by the CNN MesoNet, specialising in malignant mesothelioma diagnostics on whole slide images. As a further fascinating feature, MesoNet allows a peek into the “black box” typical to many ML algorithms, and there it can be seen that the decision of an image classification is often not based on the tumour itself, but often on the surrounding tumour and its rate of inflammation, vacuolisation and cellular diversity [65].

Interestingly, the data on the expression of genes, proteins or metabolites and its matching to a phenotype can be processed by ML in a similar way as is expressed in the above-mentioned images, although categorically different to the human eye. Since the rise of next generation sequencing and microarray techniques databases are growing, it is a major task for statistical analysis to mine for patterns that present hints for the present or future development of diseases. Yet, ML is already outperforming the human eye when it comes to the identification of the primary tumour from the data collected from a distant metastasis. A deep learning classifier was trained to predict with 91% accuracy the cancer type, based on patterns of somatic passenger mutations detected in the whole genome sequencing of 2606 tumours from 24 common cancer types [53]. A different approach maps the high-dimensional omics input data into a latent space with lower dimensionality. Based on these generated embeddings, several downstream task (including tumour type classification and survival prediction) modules are trained with a multi-task strategy, which has a higher performance than training each task individually [55].

The initiative around a tool called DeepCC goes even further, which performs the molecular subtyping of tumours and hence enables classification on its gene expression signature. In a supervised biological knowledge-based framework, a DNN creates a functional spectrum to classify colorectal cancers [54].

Prognosis prediction is one central aspect of clinical diagnosis, staging and the consequent disease management. Understanding the long-term effects of the fluctuations in gene expression and the changes of molecular pathways allows an assessment of prognosis. The freely available tool Cox-nnet represents a DNN-based algorithm that computes high throughput transcriptomics data from patients to enable an accurate and highly efficient prognosis prediction [60]. The Cox proportional hazard DNN is also employed in DeepSurv, an ML algorithm that is trained on simulated and real survival data. It models the correlations between the covariates of a patient and their risk of failure. Furthermore, the models integrate potential treatment options and correct the survival data to the success of the treatment regimen [61]. In [63], autoencoders are used to integrate the unique modalities per omics data. They tested their approach on multi-omics data for breast cancer survival prediction and reported a C-index of 0.641. Focusing on the prognosis for lung cancer, the approach in [62] uses unsupervised learning techniques to detect subtypes in non-small lung cancer, which are associated with the survival of patients and divide them into groups of longer and shorter-surviving. Another approach combines 15 extracted biomarkers with clinical data to predict the 5-year survival status of patients with non-small lung cancer. Their integrative model, which combines two DNNs, one for microarray data and the other for clinical data, achieves an AUC of 0.8163 [64].

However, not only is the analysis of transcriptomics data is time and resource consuming, but also the building of entire sequences from the up-to-1 billion reads provided from a state-of-the-art sequencer is laborious and time consuming. A specialised CNN, DeepVariant, was created to support the diagnostic facilities to faithfully reconstruct the entire genomes from the machine generated reads [68].

Next to the building and understanding of the transcriptomics data, some convenient CNNs were developed to support the researchers in designing projects. ML algorithms, such as DeepBIND, DeepChrome and DeepHistone, are tools to predict the RNA and DNA binding sites for proteins, the prediction of gene expression or prediction of histone modifications, respectively [69,70,71].

## 4. A Practical Perspective on the Challenges and Limitations

The versatile applicability of ML-based diagnostic approaches is tempting and sparked an ongoing debate about whether or when data-driven diagnostic approaches will replace traditional diagnostic approaches in the near future. Among the top percentile of the most-cited nature publications of the year 2017, for example, is the work by Esteva and colleagues, who pitched a CNN against an international panel of fully qualified dermatologists to comparatively assess their diagnostic accuracy in detecting malignant skin lesions [72]. Although the seemingly immediate clinical applicability of this and comparable diagnostic approaches is tempting, several relevant challenges and limitations needs to be kept in mind when discussing the current clinical value of ML-based diagnostic approaches.

### 4.1. Data-Driven Approaches and Human Interaction Complement Each Other

A lacking mutual understanding for the needs of clinicians and other practitioners on the one hand, and the abilities that data-driven diagnostic approaches possess today on the other hand, has led to a protective and sceptical attitude towards what the technological advances of the past decade can contribute to the current mode of clinical practice. After early attempts towards the automated detection of potentially malignant findings on mammograms and other radiographic images only led to higher recall rates, but did not yield better outcomes for the individual patient, for example, many practitioners deemed data-driven diagnostic aids unfit for the clinical practice [73]. As unreliable and faulty reports from a data-driven diagnostic approach do not only not help the practitioner, but also increase the workload and contribute to alert fatigue [70], many practitioners are hesitant to embrace the advantages that data-driven diagnostic approaches can, eventually, yield. It has, for example, been shown that integrated and collaborative (“AI-assisted”) diagnostic approaches can significantly increase the diagnostic accuracy for diabetic retinopathy, as compared with the clinical judgment or data-driven diagnostic approach alone [74].

### 4.2. Dataset Shift and the Inhomogeneity of Patient Populations

For a comprehensive applicability in clinical practice, any data-driven diagnostic approach must account for the fact that input data are generated in a dynamic, non-stationary environment. In this context, input datasets can shift significantly [75] and a data-driven approach that relies inherently on the generalisation, through interpolation and extrapolation, of the input dataset that it has been trained on, can consequently come to false conclusions [76]. While practitioners can subconsciously adapt to these dynamic changes, data-driven diagnostic approaches would require close monitoring and periodical updates to perform accordingly in a real-world context. Closely related to this issue are the concepts of brittleness and bias, which corrupt the input datasets in a way that leads to their faulty procession and false conclusions, respectively [77,78].

### 4.3. The Clinical Applicability of Accuracy Metrics Is Limited

The accuracy of most data-driven diagnostic approaches is, due to its nature in the scientific field of data science, often reported in the form of a receiver operator curve (ROC), or sensitivity and specificity, if a continuous variable is dichotomizsed. However, the former more than the latter does not necessarily translate into a practical conclusion about whether a given application provides any true benefit for the individual patient [79]. Efforts have been made to establish a more standardised approach for reporting the results from a data-based diagnostic or therapeutic approach, as, for example, the TRIPOD statement illustrates [80]. However, to fully address this challenge, a better and mutual understanding of a clinically relevant outcome and how a methodologically sound data-driven approach can lead to such needs to be established. Closing this so-called “AI chasm”, therefore, remains a priority to introduce data-driven diagnostics into the clinical practice [81]. This also includes efforts to increase the “data science literacy” among practitioners in the medical field.

### 4.4. Prospective Evaluation and Peer-Reviewed Reporting

Although the methodological sophistication of many data-driven diagnostic approaches facilitates a broad applicability in principle, the validation of these models has been mostly based on retrospective data. Esteva and colleagues, for example, based their evaluation on a total of 129,450 retrospectively collected clinical images [72], but failed to provide any subsequent prospective validation of their approach. Yet, similar to pharmaceutics and medical devices, regulatory bodies and practitioners alike require, or at least strongly prefer, a thorough prospective evaluation. As long as the majority of data-driven diagnostic approaches fails to provide prospective evaluations, market access, as well as the acceptance among practitioners, is likely to remain low. An additional important aspect focuses on where the reporting of these evaluations is published. The accustomed manner of data scientists who report their findings for an academic audience is to upload their results to pre-print servers. Although the medical field could potentially learn from this more progressive way of reporting research findings, practitioners are typically sceptical about non-peer-reviewed research. Additionally, the establishment of reporting standards and guidelines is much more difficult in the less regulated field of pre-print reports [80,82].

### 4.5. Safety and Protection from Adversarial Attacks

As a diagnostic tool, data-driven approaches need to meet safety and security standards, just as any other information technology does. To our knowledge, however, no application has been optimised to effectively resist an adversarial attack. Yet, at the same time, reports about brittleness, as outlined above, or the intentional deception of a data-driven diagnostic approach become more frequent [77,83,84]. More prominently, Finlayson and colleagues provided examples of how these fraudulent attacks lead to tangible and disadvantageous outcomes for the affected patients [85]. Although the first autonomous AI-based diagnostic tool received FDA approval [86], the debate about how developers and manufacturers of data-driven diagnostic and therapeutic approaches can make sure to meet the regulatory standards is ongoing [87]. These challenges have, from our perspective, not been addressed thoroughly enough.

### 4.6. Ethical Implications for ML in Health Care

The use of individual electronic medical records in automated big data processing creates an ethical, legal and moral conundrum. A complete medical record contains all the information to undoubtedly identify the individual. According to Western legislation, this data record always belongs to the patient and can only be accessed and analysed after obtaining a voluntary and informed consent. Yet, to train ML algorithms sufficiently, large amounts of data are required. To circumnavigate this issue, researchers started to create anonymised medical record data repositories and share them to train the ML algorithms for different applications. Such anonymised datasets are then provided by many centres and processed by researchers around the globe. However, studies show that many of these anonymisation processes are incomplete and merely produce pseudonymisation, with the potential to still withdraw information about the specific individual [88].

Not only does the data itself cause ethical and legal issues, but also the conclusions drawn out of ML generated information has to be handled with care. According to the EU legislation, each patient has the right to fully understand how the information was created that leads to his diagnosis and treatment regimen [89]. However, as mentioned above, the output created by a well-trained ML algorithm cannot be fully understood by the human mind. ML creates a highly complex analytic funnel that neither the data scientist nor the practitioner can explicitly explain. This autonomy that is given to software needs to be reflected in the regulatory framework in which these approaches work. The FDA is aware of this issue and that is the main reason why currently all approved ML apparatus have to be labelled as “diagnostic support tool” and are not allowed to be marketed as automatic diagnostic finding equipment: the intuition and experience of the practitioner still needs to remain as the final frontier of a treatment decision. This is also of crucial importance when it comes to technical limitations. ML, with all its fascinating possibilities and results, still remains a statistical application and, hence, relies on probabilistic determination. Yet, any statistical application can only be as precise as the quality of the input data. Hitherto, it is known that training sets are often incomplete, noisy and have an inherent bias that causes a legal, moral and ethical issue. Therefore, the reliability of any conclusions drawn for the clinical practice need to be handled cautiously. ML is often designed at large university centres in the developed world and thus many training data sets originate from the geographical location of the research institute. Furthermore, the creation of digital data is very cost intensive. Hence, the large accumulation of datasets is limited to wealthy health systems or to health systems that are paid for by the patient directly. The emerging precision medicine is very cost intensive and many publicly funded medical systems cannot afford this. This implies that, only from financially strong patients, digital data, such as sequencing, high resolution imaging and MRI scans, are performed and those data are fed into the ML algorithms. This again creates a bias of data available to ML based on patient selection by race, ability, sex and class.

Many of these above-mentioned issues can be combined in the emerging topic of cyber and bio-security [90]. Biological data are nowadays mostly stored as digital entities and this is also the foundation of all subsequent ML processing. However, by shifting biological and medical data into cyberspace, new threats for the integrity of the information arise. Next to all the established medical ethical standards, further standards that are in use in cybersecurity now need to be taken into account. The term cyberbiosecurity was coined in 2018, but was not known to a broad audience until the United Nations led the G7 Global Partnership against the Spread of Weapons and Materials of Mass Destruction (WMD) identified it as a key developmental field of its future activities [91].

Even if the benefits of new methods, such as the use of ML in medicine, outweigh the downsides by magnitudes, an honest and open discussion of such scenarios needs to be initiated, to create a legal and ethical safe space in which scientists can further develop the tools.

## 5. Conclusions

Non-communicable diseases in general and cancer in particular, have been shown to contribute greatly to the global burden of disease. A key factor to reduce the burden of disease from cancer is to improve the screening methods and detection models of such conditions. The status quo of the conventional approaches in this field, however, is being challenged. Specialised ML models are becoming an applicable aid along the diagnostic pathways of more and more malignant conditions. Yet, it needs to be kept in mind that ML should not be considered as a standalone diagnostic tool that can operate independently, as long as it is provided with suitable input data. Identifying and compiling suitable training data within an ethical and regulatory framework to ensure data safety and security, reduce the biases and limit overfitting that is becoming a new and required skill set for researchers. Yet, if practitioners and scientists are aware of these challenges and address them responsibly, ML yields the potential to become another key driver to address the global burden of disease from cancer.

## Figures and Tables

**Figure 1 cancers-14-00623-f001:**
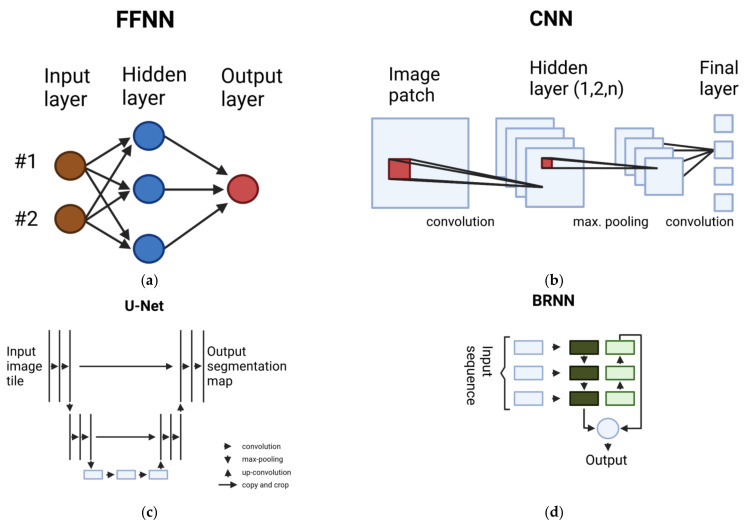
Overview of the machine learning approaches employed in medical diagnosis. (**a**) Visualisation of a feed-forward neural network with one hidden layer. (**b**) A CNN architecture with convolution and max-pooling operations ending in a fully connected final layer. (**c**) The simplified U-Net model with its characteristic shape generates a segmentation map from an input image. (**d**) In a bidirectional RNN, the input sequence is processed in both directions. Images modified from [19,20,21].

**Table 1 cancers-14-00623-t001:** Best suited machine learning algorithms for the various approaches of cancer identification. Depending on the task and the type of data, specific deep learning methods prove to be well suited.

Task	Type of Data	ML Method	Disease Spectrum	References
Segmentation	MRI images	3D-CNN, CNN	Brain tumour, prostate cancer	[37,38]
	Mammograms	CNN	Breast cancer	[39,40,41]
	Ultrasound images	U-Net (FCN)	Breast cancer	[42]
Classification	Histological images	CNN, CRNN	Breast cancer, colorectal cancer	[43,44,45]
	Dermoscopic segmentation	CNN	Skin lesions	[46,47]
	Ultrasound images	CNN	Breast cancer	[48,49]
	(Volumetric) CT scans, slides	CNN	Lung cancer	[50,51,52]
	OMICs, multi-OMICs	DNN	Various	[53,54,55]
	H&E images, slides	CNN	Colorectal cancer	[56]
	MRI images	CNN, CRNN	Brain tumour	[19,57]
	Mammograms	CNN	Breast cancer	[58]
Detection	Mammograms	CNN	Breast cancer	[58]
	CT scans	3D-CNN	Lung cancer	[59]
Prognostic	OMICs, multi-OMICs	DNN	Various	[55,60,61,62,63,64]
	Histological images	CNN	Soft tissue cancers	[65]

Abbreviations: CT abbreviations: CT—computed tomography; CNN—convolutional neural network; DNN—deep neural network; H&E—haematoxylin and eosin stain; MRI—magnetic resonance imaging; CRNN—convolutional recurrent neural networks; FCN—fully convolutional network and DL—deep learning.

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
