# Peer review of "Advances in and the Applicability of Machine Learning-Based Screening and Early Detection Approaches for Cancer: A Primer"

_cancers, 2022, doi:10.3390/cancers14030623_

Round 1

Reviewer 1 Report

This is a highly informative review and I think this is quite useful for the researchers in providing a lot of information. I would recommend it for acceptance however before that I would encourage author to incorporate few of my suggestion to improve the present manuscript.
Cancer is a complex disease and to understand the complexity, it is imperative to take an integrative and holistic approach that combines multi-omics data to dissect and understand the interrelationships of the involved molecules. It also helps in assessing the flow of information from one omics to the other and helps in bridging the gap among them. I would suggest author to incorporate a section and discuss the integration of multi-omics data in cancer diagnosis and also discuss the importance of integrating multi-omics data over single omics data analysis and how these multi-omics approaches can improve the diagnosis process.

Author Response

Our answer: Thank you to the reviewer for this encouraging words. The reviewer is absolutely right when he states that cancer is a complex disease and it is important to see the entire picture and all the available data that are generated from (prospective) patients. We were inserting now some information on multi-omics approaches in the text and are discussing it in its applicability [1]. However, we still mainly focus on the analysis of individual omics data, since it important to us, that the reader understands how data need to be designed and sorted, before a ML algorithm can compute it to reasonable results. If one fails to prepare clean input data one faces a common mathematical problem of rubbish in – rubbish out.

1. Lai, Y.-H.; Chen, W.-N.; Hsu, T.-C.; Lin, C.; Tsao, Y.; Wu, S. Overall survival prediction of nonsmall
cell lung cancer by integrating microarray and clinical data with deep learning. Scientific
Reports 2020, 10, 4679, doi:10.1038/s41598-020-61588-w.

Reviewer 2 Report

In this “Review Article”, Benning et al. aimed to depict the “state-of-the-art” in the field of machine-learning- (ML)-based algorithms potentially useful in oncology. More in detail, the authors focused on preliminary data suggesting the applicability of ML as a novel advanced tool for the screening and the early diagnosis of several types of “cancers”. Concerning all the relevant implications of this cutting-edge technological approach in medicine, the authors also focused on the drawbacks shaded by ML-based analysis. Finally, the authors emphasized the needs for standardized procedures to be applied in ML-based diagnosis and follow-up of cancer.

I think that this Review Article would be potentially of interest since it analyses a relevant issue that deserves greater attention by engineers and clinicians for the prospects of future large-scale applicability. However, I also have minor concerns that should be addressed by the authors in the attempt to improve the overall quality of the Review.

  1. I would suggest to the authors to divide the paragraph entitled "Machine learning applications in Cancer Diagnostics" in subsections reporting in more details all the relevant issues concerning the screening; early diagnosis; monitoring of patients undergoing therapeutic procedures; follow-up. Also, it would be useful to report current ML-based approaches in oncology by considering single cancers (i.e. lung cancer etc...). 
  2. I would also suggest to report information on ML-based algorithms in oncology based on technical devices used (i.e. MRI, blood analysis etc...).
  3. The Introduction on technical issues concerning ML-based analysis is full of detail and thus it is unbalanced as compared to other sections of the manuscript (i.e. ML application in cancer diagnosis).
  4. It is unclear whether the algorithms reported can be applied for the detection of non-cancerous oncological disorders.
  5. Add a comment in discussion about the telemedicine applications of ML-based devices for the monitoring of patients.
  6. Comment the applicability of ML-based algorithms in oncology for large scale use in terms of Public Health.
  7. A relevant aspect in oncology is represented by palliative care for terminal patients. How would ML be applied in this relevant field, in order to improve the quality of life of these patients?
  8. Minor spelling typos should be corrected.

Author Response

Reviewer 2
In this “Review Article”, Benning and Peintner et al. aimed to depict the “state-of-theart”
in the field of machine-learning- (ML)-based algorithms potentially useful in
oncology. More in detail, the authors focused on preliminary data suggesting the
applicability of ML as a novel advanced tool for the screening and the early diagnosis
of several types of “cancers”. Concerning all the relevant implications of this cuttingedge
technological approach in medicine, the authors also focused on the drawbacks
shaded by ML-based analysis. Finally, the authors emphasized the needs for
standardized procedures to be applied in ML-based diagnosis and follow-up of
cancer.
I think that this Review Article would be potentially of interest since it analyses a
relevant issue that deserves greater attention by engineers and clinicians for the
prospects of future large-scale applicability. However, I also have minor concerns
that should be addressed by the authors in the attempt to improve the overall quality
of the Review.
1. I would suggest to the authors to divide the paragraph entitled "Machine
learning applications in Cancer Diagnostics" in subsections reporting in more
details all the relevant issues concerning the screening; early diagnosis;
monitoring of patients undergoing therapeutic procedures; follow-up. Also, it
would be useful to report current ML-based approaches in oncology by
considering single cancers (i.e. lung cancer etc...)
Our response: Thank you for this remark. While writing this review we enjoyed the act of merging two operationally completely different professions – medicine and informatics. After long discussions, we identified the medical professional as well as the wet lab biologist as the target audience of this review – not the patient! That’s why we structured the review in a way that shows how data generated while making diagnosis can be applied to ML. Since data generated while early diagnosis, monitoring and follow up are usually of similar type we refrained from listing ML for each of these steps, since this would have been reiterating.
Nevertheless, we put a focus on ML approaches in single cancers and listed some of them in Table 1. We rephrased the paragraphs referring to the table to strengthen the link to single cancers [1-4].

2. I would also suggest to report information on ML-based algorithms in oncology
based on technical devices used (i.e. MRI, blood analysis etc...).
Our Answer: Thank you for this comment. We indeed mention the analysis of
MRI and CT scans by the use of image analysis tools. However, we
highlighted those applications even more in the text. Regarding the screening
of Blood analysis, to our knowledge, there is no publication out that was using
ML to process standard blood tests. Maybe scientist were a little reluctant to
use blood parameters in ML due to the apex of the Theranos scandal recently
– purely speculative… However, blood is often used for omics analysis (RNA
expression, Protein expression, etc.) and we covered this field quite
extensively.

3. The Introduction on technical issues concerning ML-based analysis is full of
detail and thus it is unbalanced as compared to other sections of the
manuscript (i.e. ML application in cancer diagnosis).
Our response: Thank you for this remark. Indeed we had a lot of discussion
among the authors while writing the text and it is also interesting to see that
among the reviewers (see Reviewer#3) this balance is a matter of debate. We
then decided to leave the technical and mathematical part in the review, since
we think that it is important that everyone who considers to use ML in his
patient care or scientific work should have an idea about the general concept
and mathematical approaches. ML in general has the issue of creating a
“Black box”, where the decisions are made. This is a major drawback, since
especially EU legislation enforces a closed-loop traceability of the decisionmaking.
Furthermore, having a general idea about the mathematical approach helps
the researcher to design research questions that can then be supported by
ML.

4. It is unclear whether the algorithms reported can be applied for the detection
of non-cancerous oncological disorders.
Our response: Thank you for this remark. Indeed we were a little imprecise in
this regard and tried to rephrase some of the sentences. However, the basic
concept of ML allows to train the algorithm to identify everything the doctor is
interested in, be it benign or malingnant tumors. It all depends on the
prediction task and the quality and the quantity of data set.

5. Add a comment in discussion about the telemedicine applications of ML-based devices for the monitoring of patients.
Our response: Thank you for this interesting remark. Indeed, telemedicine and
ML in theory easy go hand in hand, since both rest on the exchange of digital
data. Omics data or histological images can be sent by mail and collected all
over the globe to analyse them using ML. However, as we raised in the last
chapter, the uncontrolled distribution of digital personal medical data raises a
lot of legal and ethical concerns. At least according to EU legislations, all data
belong to the patient and the patient needs to be able at all time to know, what
happens to the data. Hence, only fully anonymised data are allowed to be sent
to other institutes – contrary to the idea of classical telemedicine.

6. Comment the applicability of ML-based algorithms in oncology for large scale
use in terms of Public Health.
Our answer: Thank you also for this interesting point that we covered very
carefully in the last outlook paragraph. Again it is, similar as to item #5 a
legislative and ethical issue. ML is not limited by any amount of data. In theory
all patients globally can be analysed in one single ML. Approaches like that
are already performed or planned, however, major focus needs to rest on
pseudonymisation of all used datasets.
Furthermore, large scale approaches in ML will currently have a strong bias to
upper-class Western patients that can afford a contemporary broad spectrum
diagnosis of their diseases. Hence, results by ML on such data needs to be
handled with care when Public Health measures are derived from them.

7. A relevant aspect in oncology is represented by palliative care for terminal
patients. How would ML be applied in this relevant field, in order to improve
the quality of life of these patients?
Our response: thank you for raising this important issue. We frear that ML will
not be of any benefit for patients already in palliative care, since usually the
amount of regular diagnostic procedures on patients in palliative care is
strongly reduced and no new data for ML are produced.
The only potential benefit for “therapied-out” patients would be that a ML can
be trained to identify such situations early to avoid that the patient has to go
unnecessary extra treatments that won’t have any benefits. To our knowledge
there are no reports on this issue and that’s why we are hesitant to elaborate
on this issue in this review.

8. Minor spelling typos should be corrected.
Our answer: Thank you and we are sorry for this sluttery. We proofread the entire
piece and got it checked by a native speaker.

Literature:

1. Lai, Y.-H.; Chen, W.-N.; Hsu, T.-C.; Lin, C.; Tsao, Y.; Wu, S. Overall survival prediction of nonsmall
cell lung cancer by integrating microarray and clinical data with deep learning. Scientific
Reports 2020, 10, 4679, doi:10.1038/s41598-020-61588-w.
2. Tong, L.; Mitchel, J.; Chatlin, K.; Wang, M.D. Deep learning based feature-level integration of
multi-omics data for breast cancer patients survival analysis. BMC Medical Informatics and
Decision Making 2020, 20, 225, doi:10.1186/s12911-020-01225-8.
3. Zhang, X.; Xing, Y.; Sun, K.; Guo, Y. OmiEmbed: A Unified Multi-Task Deep Learning
Framework for Multi-Omics Data. Cancers 2021, 13, 3047.
4. Takahashi, S.; Asada, K.; Takasawa, K.; Shimoyama, R.; Sakai, A.; Bolatkan, A.; Shinkai, N.;
Kobayashi, K.; Komatsu, M.; Kaneko, S.; et al. Predicting Deep Learning Based Multi-Omics
Parallel Integration Survival Subtypes in Lung Cancer Using Reverse Phase Protein Array Data.
Biomolecules 2020, 10, 1460.
5. Krizhevsky, A.; Sutskever, I.; Hinton,

Reviewer 3 Report

Introduction:

I would suggest that the Introduction includes the aim of the study and a very brief overview of the manuscript’s organization.

“In the first phase, the model estimates unknown dependencies in the given data, followed by the usage of these learned dependencies to predict new outputs [14]”. The second phase is not clear here. Please rephrase

A brief Introduction to ML:

Please add references to AlexNet and ResNet

Machine learning applications in Cancer Diagnostics

Table 1 contains very few examples and is only about CNNs. However, several approaches use hybrid methods which use both CNNs for image data and ML algorithms to process clinical data either independently or simultaneously. Also, many cancer types are missing (such as lung cancer). Also, the PET technology system, which is closely related to cancer malignancy rating identification, is not mentioned

On page 7 the authors introduce the concept of transfer learning. Since transfer learning is one of the most dominant CNN strategies, I suggest that this introduction would be part of section 2.

Key findings and results from the most notable studies reported in Section 3 are missing.

Since the title of the article talks about advances in ml-based detection of cancer, it would be wise to include more works from 2020 and 2021, given that they introduce something new, of course. 

A practical perspective on challenges and limitations

A common issue of DL methods is that of explainability. I believe that it should be included in Section 4.  

Author Response

Reviewer 3
Introduction:
I would suggest that the Introduction includes the aim of the study and a very brief
overview of the manuscript’s organization.
Our answer: Thank you for this remark. We included an overview- and organisation
paragraph in the introduction.
“In the first phase, the model estimates unknown dependencies in the given data,
followed by the usage of these learned dependencies to predict new outputs [14]”.
The second phase is not clear here. Please rephrase
Our answer: Sorry for this grammatical impreciseness. We rearranged the wording
and hope the concept is now clearer: “In the first phase, the model estimates
unknown dependencies in the given data, followed by the second phase that uses
these learned dependencies to predict new outputs [13]”.
A brief Introduction to ML:
Please add references to AlexNet and ResNet
Thank you for pointing out this flaw. The references were lost while building the paper
template. References were now inserted: [5,6]
Machine learning applications in Cancer Diagnostics
Table 1 contains very few examples and is only about CNNs. However, several
approaches use hybrid methods which use both CNNs for image data and ML
algorithms to process clinical data either independently or simultaneously. Also,
many cancer types are missing (such as lung cancer). Also, the PET technology
system, which is closely related to cancer malignancy rating identification, is not
mentioned
Our response: Indeed CNN is overproportionally mentioned in this review, but this
algorithm developed into the workhorse of medical ML analysis. Often CNNs are now
also employed as a subroutine in a combination of various ML algorithms, an
approach we admittedly covered faintly in the first submission. We now expanded the
collection for a report about “omniembed” that harbours interesting features for future
research [3].
Regarding the second part of the reviewer’s criticism on Table 1 we indeed already
mention lung cancer and PET scans. Still we added now more primary literature
focussing on lung cancer. Regarding PET scans, the final readout of such an
analysis is again a visual image that can be analysed by ML similar to images from
histology or similar.
On page 7 the authors introduce the concept of transfer learning. Since transfer
learning is one of the most dominant CNN strategies, I suggest that this introduction would be part of section 2.
Our answer: Thank you for this remark. We agree with the reviewer, that it eases the understanding of the reader. We transferred the introduction of CNN to section 2.

Key findings and results from the most notable studies reported in Section 3 are
missing.
Thank you for raising this issue. We are aware, that not every paper listed in our
review got the same detail of reporting. We agreed on this trade-off to enhance the legibility to the lay reader. However, if the reviewer insists, that we go into greater details with specific papers, we are happy to follow his suggestions.
Since the title of the article talks about advances in ml-based detection of cancer, it would be wise to include more works from 2020 and 2021, given that they introduce something new, of course.
Our response: Thank you for this comment. We added some more works from the recent years and have now in total 20 quotes published in 2020/21. Especially the aforementioned work on “Omniembed” caught our special attention.
A practical perspective on challenges and limitations A common issue of DL methods is that of explainability. I believe that it should be included in Section 4.
Our answer: Thank you for this remark. We hid the explainability-issue in the term “Blackbox” that we discuss in the last section of the review. However, we rephrased this paragraph to increase comprehensibility.

Round 2

Reviewer 3 Report

The authors have addressed my comments. I recommend this article for publication.